# Prevalence of Torque Teno Virus (TTV) in Cervical Precursor Lesions and Cancer in Chilean Women

**DOI:** 10.3390/ijms262211039

**Published:** 2025-11-14

**Authors:** Matías Guzmán-Venegas, Carolina Moreno-León, Cristian Andrade-Madrigal, Alejandra Román, Rancés Blanco, Iván Gallegos, Francisco Aguayo

**Affiliations:** 1Laboratorio de Oncovirología, Departamento de Ciencias Biomédicas, Facultad de Medicina, Universidad de Tarapacá, Arica 1000000, Chile; maguzmanvenegas@gmail.com (M.G.-V.); carolinajohanamoreno@gmail.com (C.M.-L.); cristian.andradem@gmail.com (C.A.-M.); 2Departamento de Obstetricia, Facultad de Ciencias de la Salud, Universidad de Tarapacá, Arica 1000000, Chile; aaromanl@academicos.uta.cl; 3Independent Researcher, Av. Vicuña Mackenna Poniente 6315, La Florida 8240000, Chile; rancesblanco1976@gmail.com; 4Departamento Anatomía Patológica, Hospital Clínico Universidad de Chile, Santiago 8380000, Chile; igallegos@hcuch.cl

**Keywords:** cervical, cancer, Torque teno virus, lesions

## Abstract

Torque teno virus (TTV) is a highly prevalent DNA virus in humans, but its role in carcinogenesis is not well understood. While human papillomavirus (HPV) is a well-established etiological agent in cervical cancer, co-infections with other viruses such as Epstein–Barr virus (EBV) or TTV may influence disease progression. We conducted a cross-sectional study using 94 formalin-fixed, paraffin-embedded (FFPE) cervical tissue samples. These specimens were collected from women with cervical intraepithelial lesions (CINI-III) or squamous cell carcinoma (SCC) at the Clinical Hospital of the University of Chile. After extracting DNA, we screened for TTV using real-time polymerase chain reaction (qPCR). Statistical analysis was performed using Fisher’s exact test. Of the 94 samples, 83 were positive for the human β-globin gene and included in the final analysis. TTV was detected in 12.0% (10/83) of these samples. Among the TTV-positive cases, the virus was most frequently detected in high-grade lesions (70.0%), followed by low-grade lesions (20.0%) and squamous cell carcinoma (10.0%). However, these differences were not statistically significant (*p* = 0.688). This is the first study to assess TTV prevalence in cervical lesions among Chilean women. Although we found no statistically significant associations, a higher frequency of TTV was detected in precursor lesions compared to SCC. Further studies are needed to understand the potential immunomodulatory role of TTV in cervical carcinogenesis.

## 1. Introduction

Cervical cancer remains one of the most prevalent malignancies among women worldwide, particularly in low- and middle-income countries. In Chile, it ranks as the third most common cancer among women, imposing a significant burden of morbidity and mortality [1]. The development of cervical cancer typically follows a multistep progression from low-grade squamous intraepithelial lesions (LSIL) to high-grade squamous intraepithelial lesions (HSIL), culminating in invasive squamous cell carcinoma (SCC) or adenocarcinoma. Histopathologically, these stages are classified as cervical intraepithelial neoplasia grades 1 through 3 (CINI-III), with CINIII carrying the highest risk of progression to invasive cancer [2]. Persistent infection with high-risk human papillomavirus (HR-HPV), particularly genotypes 16, 18, 31, 33, and 35, among others, is a necessary etiological factor in nearly 100% of cervical cancer cases [3]. However, HPV infection alone is insufficient to cause malignancy, as most infections are spontaneously cleared by the immune system [4]. Cofactors such as early sexual initiation, multiple sexual partners, smoking, low socioeconomic status, and immunosuppression have been associated with persistence of HPV infection and cancer progression [5,6,7]. Additionally, other viral agents, such as Epstein–Barr virus (EBV) and cytomegalovirus (CMV), have been proposed as potential contributors to cervical carcinogenesis, although the evidence remains controversial [7,8].

Torque teno virus (TTV), is a member of the Anelloviridae family, that is highly prevalent in the human population and is often detected in asymptomatic individuals [9,10,11]. Indeed, viruses of the Anelloviridae family are small (30–32 nm in diameter), non-enveloped virions characterized by T = 1 icosahedral symmetry and a circular, negative-sense single-stranded DNA genome [12]. The anellovirus genome utilizes a rolling-circle replication mechanism within the host nucleus, with genome length varying by genera (e.g., TTV: 3.5–3.9 Kb). This process is facilitated by a conserved, GC-rich non-coding region. The coding region contains 3 to 5 main overlapping open reading frames (ORFs) that employ alternative splicing to produce 6 to 7 different proteins (ranging from 12 to 80 kDa). Notably, ORF1 encodes the largest of these—the highly variable capsid protein—which is used for classification and aids in immune evasion. Other proteins, such as those from ORF2 and ORF3, are implicated in manipulating host functions, including suppressing immune pathways. TTV belongs to the *alphatorquevirus* genus with 29 species [13]. The TTV genome exhibits a stark contrast in variability between its regions: the non-coding region (UTR) is highly conserved, while the coding regions (particularly ORF1) are hypervariable [14,15]. This dichotomy is critical for diagnostics as the conserved UTR is the ideal target for reliable PCR detection of all TTV variants, whereas early studies targeting variable ORF1 regions significantly underestimated viral prevalence [16,17].

Originally identified in the context of post-transfusion hepatitis of unknown etiology, TTV has since been found in a wide range of tissues, including liver, blood, respiratory tract, and various tumors such as breast, lung, and gastrointestinal cancers [18,19,20,21]. Its role in oncogenesis is not fully understood, but evidence suggests that TTV may influence immune modulation through mechanisms such as suppression of the NF-κB pathway and evasion of immune detection via hypervariable genomic regions and viral microRNAs [22,23,24,25]. TTV has also been detected in cervical samples, including in patients with or without HPV infection [26,27,28]. A recent study suggests that TTV may be more prevalent in CINI and SCC, particularly in the presence of HR-HPV [26]. Furthermore, the co-presence of EBV and TTV in cervical lesions has prompted speculation about their potential cooperative effects on immune evasion or oncogenic transformation [29,30,31]. Despite growing interest, the epidemiological and biological role of TTV in cervical cancer remains poorly characterized, especially in Latin American populations. To date, no studies have evaluated TTV prevalence in cervical lesions from Chilean women. This study aims to determine the prevalence of TTV in paraffin-embedded cervical tissue samples representing various stages of disease progression.

## 2. Results

### 2.1. Clinicopathological Characteristics of the Study Cohort

In this study, 94 cervical lesions and carcinomas were analyzed for TTV presence. Eleven samples were excluded because they were negative for amplification of a fragment of the human β-globin gene. Therefore, 83 (88.3%) were positive for the constitutive β-globin gene (20–58 years old). Of the group of patients analyzed in this study, high-grade lesions and SCC are more common in the older age group, as expected. Most of the squamous cell carcinoma cases analyzed were moderately differentiated. Table 1 describes the characteristics of the Chilean women included in the study according to the grade of lesion (low or high grade and squamous cell carcinoma) according to the age range of 20–30 years (youth) and 31–58 years (adulthood). A significant association was observed between patient age and lesion grade (*p* = 0.043, Fisher’s exact test), with women in the 31–58-year age group accounting for a higher proportion of high-grade lesions and squamous cell carcinoma cases. Additionally, 36.1% samples (30/83) were HPV16+ and 2.4% (2/83) were HPV18+ [32].

### 2.2. Prevalence of TTV in Cervical Lesions from Chilean Women

TTV infection was assessed by real-time PCR, using the NG 779/NG780 primers as a mixture of forward primers and NG785 as a reverse primer [33]. Intraepithelial neoplastic lesions when classified according to the degree of injury are grouped into low-grade (CIN I) and high-grade (CIN II to CIN III). TTV DNA was detected mostly in high-grade lesions, compared with low-grade lesions and squamous cell carcinoma. However, no statistically significant difference was observed (*p* = 0.688, Fisher’s exact test). The total number of TTV-positive samples was 10/83 (12.0%) (Table 2).

Subsequently, we determined the co-presence of TTV and HR-HPV in cervical lesions. The HR-HPV prevalence and genotyping was previously determined by PCR/sequencing [32]. Analysis of the co-presence of both viruses was not statistically significant (*p* = 1.000; Fisher’s exact test) between samples positive for TTV and HPV and those positive only for TTV (Table 3).

Subsequently we analyzed the co-presence of TTV and EBV in the cervical lesions. The presence of EBV was previously determined by PCR/ISH [32]. It was observed that most samples positive for TTV were also positive for EBV. However, no statistically significant association was observed between the two variables (*p* = 0.302, Fisher’s exact test) (Table 4).

## 3. Discussion

It has been suggested that TTV could act as a risk factor in the development of cervical cancer [28]. Therefore, in this study we addressed the prevalence of TTV in cervical lesions and carcinomas from Chilean women. TTV was detected in 10/83 (12.0%) cervical lesions and carcinomas. Furthermore, TTV positivity was higher in samples with intraepithelial neoplastic lesions (90.0%) than in squamous cell carcinoma (10.0%), though this difference was not statistically significant. This is the first study in Chile reporting TTV prevalence in cervical lesions and carcinomas. These results are comparable to existing evidence, in which a slightly higher positivity has been observed in samples with high-grade lesions compared to low-grade lesions, but without a statistically significant association [34]. In fact, studies reporting TTV prevalence in cervical lesions and cancer are scarce. In Latin America, TTV has previously been reported to be present in 57.1% of cervical cancer cases from Brazilian women [28]. In Europe, a study reported a significantly higher presence of TTV in cervical carcinomas from Hungary when compared to controls. However, no association was observed between cervical abnormalities and TTV genogroup 1, which differs from those detected in head and neck lesions [35]. The discrepancy between our 12.0% prevalence and the 57.1% reported in Brazilian women could be attributable to several factors, including differences in the sensitivity of molecular detection methods, the use of FFPE samples which may contain degraded DNA, or genuine geographic variations in TTV epidemiology within South America. Of note, 86.3% prevalence of TTV has been reported in different subjects, with a significant correlation with EBV DNA in whole blood [36]. Interestingly, a significant co-presence (90%) between TTV and EBV has been reported in diffuse large B-cell lymphoma, follicular lymphoma, and Hodgkin’s disease-multiple sclerosis [29], suggesting the possibility of a cooperation between both viruses. Moreover, enhanced replication of TTV has been observed in EBV-positive cells in patients with multiple sclerosis, indicating functional cooperation of both viruses in the development of the disease [37].

Regarding the mechanistic role of TTV, it has been suggested that TTV ORF2 suppresses NF-κB pathway activity through negative translocation of p50 and p65 proteins to the cell nucleus. This would affect the transcription of cellular genes encoding IL-6, IL-8, and COX-2 [23]. However, transcripts for IL-29, IL-28A, IFN-α, and IFN-β are not altered [38]. Additionally, antibody formation against peptides encoded by different TTV ORFs has been observed [39,40,41] and antibodies against TTV appear 100 to 150 days after a blood transfusion [40]. TTV has been shown to evade the humoral immune response through mutations in the P2 hypervariable region encoded by ORF1 [24]. On the other hand, a decrease in the CD4+ T cell count has been observed when the TTV concentration is higher [42]. Importantly, in this study, TTV was enriched in precursor lesions when compared with cervical carcinomas. Although our findings suggest that TTV is not directly associated with cervical cancer progression, the higher prevalence in precursor lesions could be speculatively explained by a “hit and run” mechanism. In this model, a virus might mediate initial cellular transformation before being cleared by the immune system as the cancer progresses. This could explain the higher prevalence of TTV detected in samples with precursor lesions compared to cervical cancer cases. However, the data from this study are insufficient to support this hypothesis, which would require longitudinal studies to investigate further [43]. Indeed, a hit and run mechanism has been described for HPV18 in anogenital cancer [44] or HPV genus Beta for non-melanoma skin cancers [45,46,47].

High-risk HPVs are the known etiological agent involved in cervical cancer development worldwide [48]. However, not all initially infected women develop cervical cancer as most of them naturally clear the infection, suggesting additional cofactors [49]. Thus, it has been suggested that certain viral coinfections could act as a cofactor in the development of this type of cancer [50]. However, in this study the co-presence of TTV and HPV was identified in 60% of specimens without statistical association with progression of the cervical lesions. On the other hand, one study explored the microbiome in HPV-dominated (HD) and non-HPV-dominated (NHD) cervical swab samples. TTV8, 16 and 25 were uniquely detected in non-HPV-dominated lesions, although TTV1 was exclusively detected in HPV-dominated lesions [51]. On the other hand, EBV is a highly prevalent persistent virus that has been previously proposed to be involved as cofactor in cervical cancer [52,53]. Previously, we reported that EBV was detected in this same cohort of cervical lesions and carcinomas without a statistically significant difference among them [32]. In this study, the co-presence of EBV and TTV, similarly, was not associated with progression of cervical lesions.

This study shows some limitations. First, the low number of clinical specimens analyzed. The limited sample size may have provided insufficient statistical power to detect a modest association between TTV and lesion grade. Second, formalin-fixed and paraffin-embedded tissues frequently exhibit a high DNA fragmentation. Furthermore, DNA degradation in FFPE samples could lead to an underestimation of the true viral prevalence, meaning our reported 12% frequency should be considered a conservative estimate. Future research should prioritize longitudinal cohort studies to track TTV presence and viral load over time in women with HPV infections. Furthermore, employing next-generation sequencing (NGS) could provide a more comprehensive view of the entire cervical virome, clarifying the complex interplay between TTV, HPV, and other potential viral cofactors in cervical carcinogenesis. Quantifying TTV viral load, rather than just determining its presence or absence, may also reveal dose-dependent effects on disease progression.

In summary, this study found that TTV is present in 12% of cases of precursor lesions and cervical cancer, with non-statistically significant difference among them. Additional studies including increased sample size are warranted to dissect the potential role of TTV in human cancer.

## 4. Materials and Methods

### 4.1. Clinical Specimens

This descriptive, cross-sectional study utilized a retrospective, non-probabilistic, and purposive sampling design. Ninety-four formalin-fixed, paraffin-embedded (FFPE) cervical tissue samples were obtained from the archives of the Pathological Anatomy Department at the Clinical Hospital of the University of Chile. The samples, collected between 2005 and 2010, were from women diagnosed with low-grade squamous intraepithelial lesions (LSIL), high-grade squamous intraepithelial lesions (HSIL), or squamous cell carcinoma (SCC). Histological diagnosis and classification were performed according to the Bethesda and World Health Organization (WHO) system by an experienced pathologist. The detection of HPV and EBV in these specimens was previously carried out by conventional PCR and in situ hybridization (ISH) [32]. This study was approved by the Ethics Committee of the University of Chile. All data were anonymized, and samples were handled in accordance with institutional bioethical standards.

### 4.2. DNA Extraction and Real-Time Polymerase Chain Reaction (qPCR)

Paraffin-embedded tissue was processed to obtain 10-micron sections. These sections were then deparaffinized and incubated overnight at 56 °C with a digestion buffer containing 20 mg/mL proteinase K (Fermelo Biotec, Santiago, Chile) and 5 mL of TE Buffer (ThermoFisher-Scientific, Waltham, MA, USA). After incubation, the samples were heated at 95 °C for 10 min to inactivate the proteinase K, followed by centrifugation at 14,000 rpm for 10 min to pellet the debris. The supernatant containing the extracted DNA was then collected and was directly used for PCR as previously reported [54]. The quality of the extracted DNA was assessed by amplifying the constitutive β-globin gene using real-time PCR (qPCR). The reaction mixture, prepared in a final volume of 10 µL, included 1 µL of DNA sample, 5 µL of SensiFast (SYBR Lo-Rox One-Step Kit, Bioline, Memphis, TN, USA), 3.6 µL of nuclease-free water, and 0.2 µL of each primer. The forward primer PCO3 and reverse primer PCO4 (both at 20 mM) were synthesized by Integrated DNA Technologies (IDT, Coralville, IA, USA) and obtained from Fermelo Biotec (Santiago, Chile). Primer details are provided in Table 5. The amplification protocol was as follows: initial denaturation at 95 °C for 2 min, followed by 40 cycles of 95 °C for 5 s, 60 °C for 10 s, and 72 °C for 15 s. A CFX Opus 96 Thermal Cycler (Bio-Rad, Hercules, CA, USA) was used for this protocol. A sample was considered positive for β-globin if its melting temperature (Tm) was 82.5 °C. Samples that were negative for β-globin amplification were subjected to a purification step using the EZNA Tissue DNA Kit (Omega Bio-tek, Norcross, GA, USA), following the manufacturer’s protocol. The purified DNA was quantified using a NanoDrop One spectrophotometer (ThermoFisher-Scientific, Waltham, MA, USA). Following purification, these samples were re-analyzed for β-globin gene amplification. Negative controls included a no-template control (NTC) to check for reagent contamination and a lysis buffer control to rule out contamination during the initial tissue lysis.

TTV DNA detection was performed using a semi-nested real-time PCR protocol with a CFX Opus 96 thermal cycler. Two separate reactions were performed. The first targeted anellovirus, using 2 µL of sample and 8 µL of PCR mix. The mix contained 5 µL of SensiFast SYBR Lo-Rox One-Step Kit (Bioline, Memphis, TN, USA), 2.6 µL of nuclease-free water, and 0.2 µL of each primer mix. Two primer mixes were used in a 1:1 ratio: the forward mix consisted of NG779 and NG780 (Integrated DNA Technologies, Coralville, IA, USA), and the reverse mix of NG781 and NG782 (Integrated DNA Technologies). The second reaction targeted TTV, using 1 µL of the amplified product from the first PCR and 9 µL of PCR mix containing 5 µL of SensiFast, 3.6 µL of nuclease-free water, and 0.2 µL of primers. This reaction used the same forward primer mix and the TTV-specific reverse primer NG785 (Integrated DNA Technologies, Coralville, IA, USA). The cycling conditions consisted of an initial denaturation at 95 °C for 2 min, followed by 45 cycles of 95 °C for 5 s, 50 °C for 10 s, and 72 °C for 15 s. All PCRs, including those for β-globin, were performed in triplicate. Samples were considered positive when the melting temperature (Tm) ranged from 87.0 to 88.0 °C. The same negative controls used in the β-globin qPCR were included, and the positive control corresponded to a TTV-positive sample. PCR-amplified products were additionally characterized by agarose gel electrophoresis.

### 4.3. Statistical Analysis

Statistical analysis was performed using Fisher’s exact test (STATA v18, StataCorp LLC, College Station, TX, USA), considering *p*-values < 0.05 as statistically significant.

## 5. Conclusions

This study reports a 12% prevalence of TTV in cervical lesions from Chilean women, with higher detection in HSIL compared to SCC. Although no statistically significant associations were identified with cervical lesions. The study highlights the need for further research on the immunomodulatory role of TTV and its potential contribution to cervical carcinogenesis.

## Figures and Tables

**Table 1 ijms-26-11039-t001:** Clinicopathological features of the women included in this study.

Variables		Cervical Lesions			Total	*p*-Value
		Low Grade	High Grade	SQC		
Age range (Years-old)	20–30	8 (40.0)	11 (55.0)	1 (5.0)	20	0.043
	31–58	12 (19.0)	34 (54.0)	17 (27.0)	63	
Total age range		20	45	18	83	
Differentiation	Negative	20 (100)	45 (100)	0 (0)	65	-
	Well	0 (0)	0 (0)	2 (11)	2	
	Moderately	0 (0)	0 (0)	9 (50)	9	
	Poor	0 (0)	0 (0)	2 (11)	2	
	Unknown	0 (0)	0(0)	5 (28)	5	
Total		20	45	18	83	

**Table 2 ijms-26-11039-t002:** TTV detection in cervical lesions from Chilean women.

TTV	Cervical Lesions (%)			Total (%)	*p*-Value
	Low Grade	High Grade	Squamous Cell Carcinoma		
Positive	2 (20.0)	7 (70.0)	1 (10.0)	10 (12)	0.688
Negative	18 (24.7)	38 (52.0)	17 (23.3)	73 (88)	
Total	20	45	18	83 (100)	

**Table 3 ijms-26-11039-t003:** Frequency of TTV/HPV co-presence in cervical lesions from Chilean women.

**TTV**	**HPV (%)**		**Total**	***p*-Value**
	**Negative**	**Positive**		
Negative	29 (39.7)	44 (60.3)	73	1.000
Positive	4 (40.0)	6 (60.0)	10	
Total	33	50	83	

**Table 4 ijms-26-11039-t004:** Frequency of TTV/EBV co-presence in cervical lesion samples.

TTV	EBV (%)		Total	*p*-Value
	Negative	Positive		
Negative	31 (42.5)	42 (57.5)	73	0.302
Positive	2 (20.0)	8 (80.0)	10	
Total	33	50	83	

**Table 5 ijms-26-11039-t005:** Primers for qPCR used in this study.

Gene	Sequence 5′-3′		Tm (°C)	Amplicon (pb)
TTV UTR	Direction NG780	RGTGRCGAATGGYWGAGTTT	87–88 °C	113
	Direction NG779	ACWKMCGAATGGCTGAGTTT		
	Antisense NG785	CCCCTTGACTBCGGTGTGTAA		
Anellovirus	Direction NG779	ACWKMCGAATGGCTGAGTTT		128
	Direction NG780	RGTGRCGAATGGYWGAGTTT		
	Antisense NG781	CCCKWGCCCGARTTGCCCCT		
	Antisense NG782	AYCTWGCCCGAATTGCCCCT		
β-globin	Sense PCO3	ACACAACTGTGTTCACTAGC	82.5 °C	110
	Antisense PCO4	CAACTTCATCCACGTTCACC		

## Data Availability

The original contributions presented in this study are included in the article. Further inquiries can be directed to the corresponding author.

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
