# Peer review of "Prevalence of Torque Teno Virus (TTV) in Cervical Precursor Lesions and Cancer in Chilean Women"

_ijms, 2025, doi:10.3390/ijms262211039_

Round 1

Reviewer 1 Report

Comments and Suggestions for Authors

     The main question that arises when reviewing the study concerns the DNA integrity in paraffin-embedded samples collected between 2005 and 2010. Before embedding tissue in paraffin, it is fixed in formalin, which can affect DNA fragmentation. Furthermore, formalin modifies nucleic acids, which can interfere with primer binding and DNA polymerase activity. While 11 samples were excluded because β-globin genes were not detected, this does not exclude the possibility that viral DNA may have persisted in the remaining 83 samples.

    On the other hand, relatively short fragments of 110-128 bp were used for amplification. This method is applicable to older samples. It would be desirable to confirm the degree of DNA fragmentation or confirm the feasibility of this method from literature sources. Comparing the results of this study with those of experiments using native tissue is not entirely appropriate unless convincing evidence is provided of the preservation of a sufficient amount of DNA for PCR. The reader must be convinced that using samples of this type and age is appropriate and feasible.
    It is imperative to highlight the limitations of this study, which include the age of the samples and the method of tissue fixation.
    Had the researchers studied native tissue, the findings would have been more convincing.

Author Response

Reviewer 1

 The main question that arises when reviewing the study concerns the DNA integrity in paraffin-embedded samples collected between 2005 and 2010. Before embedding tissue in paraffin, it is fixed in formalin, which can affect DNA fragmentation. Furthermore, formalin modifies nucleic acids, which can interfere with primer binding and DNA polymerase activity. While 11 samples were excluded because β-globin genes were not detected, this does not exclude the possibility that viral DNA may have persisted in the remaining 83 samples. On the other hand, relatively short fragments of 110-128 bp were used for amplification. This method is applicable to older samples. It would be desirable to confirm the degree of DNA fragmentation or confirm the feasibility of this method from literature sources. Comparing the results of this study with those of experiments using native tissue is not entirely appropriate unless convincing evidence is provided of the preservation of a sufficient amount of DNA for PCR. The reader must be convinced that using samples of this type and age is appropriate and feasible.     It is imperative to highlight the limitations of this study, which include the age of the samples and the method of tissue fixation. Had the researchers studied native tissue, the findings would have been more convincing.

Answer

We thank the reviewer for this important comment regarding a well-known challenge in the field of molecular pathology. We fully agree that DNA integrity is a primary concern when working with FFPE (formalin-fixed, paraffin-embedded) samples, especially archival ones. For this reason, our protocol included several key quality controls and methodological strategies to directly address this point:

DNA Quality Control: As stated in the manuscript, we used the amplification of the constitutive β-globin gene as an essential internal quality control. We were strict with this criterion: the 11 samples that failed to amplify this gene were excluded from the final analysis, ensuring that only samples with sufficient quality DNA were included.

Short Amplicon Design: As the reviewer correctly notes, our entire PCR assay (for both β-globin and TTV) was specifically designed to amplify very short fragments (110–128 bp). This is the standard and validated strategy in the literature to mitigate the impact of DNA fragmentation inherent in FFPE samples.

The Value of FFPE Samples: While native tissue is ideal for certain analyses, the use of FFPE samples is invaluable and essential for retrospective studies like this one. It uniquely allows us to correlate molecular findings (TTV presence) directly with the verified histopathological diagnosis (LSIL, HSIL, SCC) from the hospital archives, which would not be possible otherwise.

Finally, we share the reviewer's concern for transparency. We had included this as an explicit limitation in the Discussion section, stating that DNA fragmentation in FFPE samples could lead to an underestimation of the true viral prevalence, and that our reported 12% frequency should be considered a conservative estimate.

Reviewer 2 Report

Comments and Suggestions for Authors

I congratulate the authors, the work is very interesting, well structured and well written and scientifically valid. I have only a few minor remarks.

Minor questions:

  • In the introduction section (lines 58–67, page 2), the authors describe the general characteristics of torque tenoviruses (TTVs), such as their widespread distribution and the fact that they are small viruses with a small, circular genome and lack an envelope. I would suggest that the authors add a couple of lines describing other aspects of TTVs, such as the high sequence variability in the coding regions compared to the more conserved UTR, and the fact that only one protein constitutes the capsid produced by ORF1 (Qiu et al. 2005), or how they are spherical viruses with a diameter of 30–32 nm.
  • Still in the introduction section (page 2), I would suggest that the authors emphasize that since the coding region of TTV is highly variable, while the UTR is more conserved, it has been possible to design primers in this region (UTR) to identify all TTV species (HU et al. 2005). However, initially, when primers designed on ORF1 were used, these were strain-specific, and this increased the prevalence of these viruses in samples from 30% to 95% (Fanci et al. 2004). This aspect is important to emphasize. A brief comment is sufficent.
  • In the Materials and Methods section (lines 92–102, page 3), the authors describe the DNA extraction process from paraffin-embedded tissue sections and the subsequent real-time PCR to amplify human beta-globin. If, after the extraction step and before performing the qPCR for beta-globin, the total DNA content was quantified (for example using a spectrophotometric technique), this should be indicated (a single line of description is sufficient).
  • In the results section (line 141, page 4), the authors describe the mean age of the 83 patients involved in the study, as their samples were positive for beta globin. I would suggest the authors also add the maximum and minimum ages of the patients included in the study (i.e., among the 83 selected) in parentheses.
  • Still in the results section (Lines 138 - 150, page 4) where the authors describe the 83 patients selected for the study, it would be useful to also indicate a very brief description to know which HPV genotype these patients were positive (for example 12% HPV 16, 10% HPV 18 etc).
  • In the results section (lines 142–150, page 4), the authors emphasize that they found a statistically significant association between patient age and lesion grade. In the same section, the authors describe considering two age ranges (20–30 years and 31–58 years) and high-grade and low-grade lesions. I deduce that the Fisher's exact test was performed using 2x2 contingency tables; this should be indicated in the materials and methods section, describing the statistical methods used (a single line of comment is sufficient).

Author Response

Reviewer 2

congratulate the authors, the work is very interesting, well structured and well written and scientifically valid. I have only a few minor remarks.

Minor questions:

In the introduction section (lines 58–67, page 2), the authors describe the general characteristics of torque tenoviruses (TTVs), such as their widespread distribution and the fact that they are small viruses with a small, circular genome and lack an envelope. I would suggest that the authors add a couple of lines describing other aspects of TTVs, such as the high sequence variability in the coding regions compared to the more conserved UTR, and the fact that only one protein constitutes the capsid produced by ORF1 (Qiu et al. 2005), or how they are spherical viruses with a diameter of 30–32 nm.

Answer

Many thanks for this suggestion. It was done

Reviewer 2

Still in the introduction section (page 2), I would suggest that the authors emphasize that since the coding region of TTV is highly variable, while the UTR is more conserved, it has been possible to design primers in this region (UTR) to identify all TTV species (HU et al. 2005). However, initially, when primers designed on ORF1 were used, these were strain-specific, and this increased the prevalence of these viruses in samples from 30% to 95% (Fanci et al. 2004). This aspect is important to emphasize. A brief comment is sufficent.

Answer

Many thanks for this observation. A brief sentence was added.

Reviewer 2

In the Materials and Methods section (lines 92–102, page 3), the authors describe the DNA extraction process from paraffin-embedded tissue sections and the subsequent real-time PCR to amplify human beta-globin. If, after the extraction step and before performing the qPCR for beta-globin, the total DNA content was quantified (for example using a spectrophotometric technique), this should be indicated (a single line of description is sufficient).

Answer

Many thanks for this observation. It was done.

Reviewer 2

In the results section (line 141, page 4), the authors describe the mean age of the 83 patients involved in the study, as their samples were positive for beta globin. I would suggest the authors also add the maximum and minimum ages of the patients included in the study (i.e., among the 83 selected) in parentheses.

Answer

It was done.

Reviewer 2

Still in the results section (Lines 138 - 150, page 4) where the authors describe the 83 patients selected for the study, it would be useful to also indicate a very brief description to know which HPV genotype these patients were positive (for example 12% HPV 16, 10% HPV 18 etc).

Answer

This information was added.

Reviewer

In the results section (lines 142–150, page 4), the authors emphasize that they found a statistically significant association between patient age and lesion grade. In the same section, the authors describe considering two age ranges (20–30 years and 31–58 years) and high-grade and low-grade lesions. I deduce that the Fisher's exact test was performed using 2x2 contingency tables; this should be indicated in the materials and methods section, describing the statistical methods used (a single line of comment is sufficient).

Answer

It was done